# SET PREDICTION WITHOUT IMPOSING STRUCTURE AS CONDITIONAL DENSITY ESTIMATION

**David W. Zhang[1], Gertjan J. Burghouts[2], Cees G. M. Snoek[1]**
[1]University of Amsterdam
{w.d.zhang, cgmsnoek}@uva.nl
[2]TNO
{gertjan.burghouts}@tno.nl

## ABSTRACT

Set prediction is about learning to predict a collection of unordered variables with unknown interrelations. Training such models with set losses imposes the structure of a metric space over sets. We focus on stochastic and underdefined cases, where an incorrectly chosen loss function leads to implausible predictions. Example tasks include conditional point-cloud reconstruction and predicting future states of molecules. In this paper, we propose an alternative to training via set losses by viewing learning as conditional density estimation. Our learning framework fits deep energy-based models and approximates the intractable likelihood with gradient-guided sampling. Furthermore, we propose a stochastically augmented prediction algorithm that enables multiple predictions, reflecting the possible variations in the target set. We empirically demonstrate on a variety of datasets the capability to learn multi-modal densities and produce different plausible predictions. Our approach is competitive with previous set prediction models on standard benchmarks. More importantly, it extends the family of addressable tasks beyond those that have unambiguous predictions.

## 1 INTRODUCTION

This paper strives for set prediction. Making multiple predictions with intricate interactions is essential in a variety of applications. Examples include predicting the set of attributes given an image (Rezatofighi et al., 2020), detecting all pedestrians in video footage (Wang et al., 2018) or predicting the future state for a group of molecules (Noé et al., 2020). Because of their unordered nature, sets constitute a challenge for both the choice of machine learning model and training objective. Models that violate permutation invariance suffer from lower performance, due to the additional difficulty of needing to learn it. Similarly, loss functions should be indifferent to permutations in both the ground-truth and predictions. Additional ambiguity in the target set exacerbates the problem of defining a suitable set loss. We propose Deep Energy-based Set Prediction (DESP) to address the permutation symmetries in both the model and loss function, with a focus on situations where multiple plausible predictions exist. DESP respects the permutation symmetry, by training a permutation invariant energy-based model with a likelihood-based objective.

In the literature, assignment-based set distances are applied as loss functions (Zhang et al., 2019; Kosiorek et al., 2020). Examples include the Chamfer loss (Fan et al., 2017) and the Hungarian loss (Kuhn, 1955). Both compare individual elements in the predicted set to their assigned ground-truth counterpart and vice-versa. While they guarantee permutation invariance, they also introduce a structure over sets, in the form of a metric space. Choosing the wrong set distance can result in implausible predictions, due to interpolations in the set space for underdefined problems. For example, Fan et al. (2017) observe different set distances to lead to trade-offs between fine-grained shape reconstruction and compactness, for 3d reconstruction from RGB images. As an additional shortcoming, optimizing for a set loss during training poses a limitation on the family of learnable data distributions. More specifically, conditional multi-modal distributions over sets *cannot* be learned by minimizing an assignment-based set loss during training. To overcome the challenges of imposed structure *and* multi-modal distributions, we propose to view set prediction as a conditional density

estimation problem, where $P(\boldsymbol{Y}|\boldsymbol{x})$ denotes the distribution for the target set $\boldsymbol{Y}$ given observed features $\boldsymbol{x}$.

In this work we focus on distributions taking the form of deep energy-based models (Ngiam et al., 2011; Zhai et al., 2016; Belanger & McCallum, 2016):

$$P_{\boldsymbol{\theta}}(\boldsymbol{Y}|\boldsymbol{x}) = \frac{1}{Z(\boldsymbol{x};\boldsymbol{\theta})} \exp\left(-E_{\boldsymbol{\theta}}(\boldsymbol{x}, \boldsymbol{Y})\right), \tag{1}$$

with $Z$ as the partition function and $E_{\boldsymbol{\theta}}$ the energy function with parameters $\boldsymbol{\theta}$. The expressiveness of neural networks (Cybenko, 1989) allows for learning multi-modal densities $P_{\boldsymbol{\theta}}(\boldsymbol{Y}|\boldsymbol{x})$. This sets the approach apart from forward-processing models, that either require conditional independence assumptions (Rezatofighi et al., 2017), or an order on the predictions, when applying the chain rule (Vinyals et al., 2016). Energy-based prediction is regarded as a non-linear combinatorial optimization problem (LeCun et al., 2006):

$$\hat{\boldsymbol{Y}} = \arg\min_{\boldsymbol{Y}} E_{\boldsymbol{\theta}}(\boldsymbol{x}, \boldsymbol{Y}), \tag{2}$$

which is typically approximated by gradient descent for deep energy-based models (Belanger & McCallum, 2016; Belanger et al., 2017). We replace the deterministic gradient descent with a stochastically augmented prediction algorithm, to account for multiple plausible predictions. We show that our stochastic version outperforms standard gradient descent for set prediction tasks.

Our main contribution is DESP, a training and prediction framework for set prediction, that removes the limitations imposed by assignment-based set losses. Sampling plays a key role in DESP. For training, sampling approximates the intractable model gradients, while during prediction, sampling introduces stochasticity. We show the generality of our framework by adapting recently proposed permutation invariant neural networks as set prediction deep energy-based models. We demonstrate that our approach (i) learns multi-modal distributions over sets (ii) makes multiple plausible predictions (iii) generalizes over different deep energy-based model architectures and (iv) is competitive even in non-stochastic settings, without requiring problem specific loss-engineering.

## 2 Deep Energy based Set Prediction

### 2.1 Training

Our goal is to train a deep energy based model for set prediction, such that all plausible sets are captured by the model. Regression models with a target in the $\mathbb{R}^d$ space, that are trained with a root mean-square error (RMSE) loss, implicitly assume a Gaussian distribution over the target. Analog to the RMSE, assignment-based set losses assume a uni-modal distribution over the set space. Training with the negative log-likelihood (NLL) circumvents the issues of assignment-based set losses. Notably, NLL does not necessitate explicit element-wise comparisons, but treats the set holistically. We reformulate the NLL for the training data distribution $P_{\mathcal{D}}$ as:

$$\mathbb{E}_{(\boldsymbol{x},\boldsymbol{Y})\sim P_{\mathcal{D}}} \left[ -\log(P_{\boldsymbol{\theta}}(\boldsymbol{Y}|\boldsymbol{x})) \right] = \mathbb{E}_{(\boldsymbol{x},\boldsymbol{Y})\sim P_{\mathcal{D}}} \left[ E_{\boldsymbol{\theta}}(\boldsymbol{x},\boldsymbol{Y}) \right] + \mathbb{E}_{\boldsymbol{x}\sim P_{\mathcal{D}}} \left[ \log(Z(\boldsymbol{x};\boldsymbol{\theta})) \right]. \tag{3}$$

The gradient of the left summand is approximated by sampling a mini-batch of $n$ tuples $\{(\boldsymbol{x}_i, \boldsymbol{Y}_i^+)\}_{i=0..n}$ from the training set. The gradient of the right summand is approximated by *solely* sampling input features $\{\boldsymbol{x}_i\}_{i=0..m}$. Directly evaluating $\frac{\partial}{\partial\boldsymbol{\theta}}\log(Z(\boldsymbol{x};\boldsymbol{\theta}))$ is intractable; instead we approximate the gradient by sampling $\{\boldsymbol{Y}_j^-\}_{j=0..k}$ from the model distribution:

$$\frac{\partial}{\partial\boldsymbol{\theta}}\log(Z(\boldsymbol{x};\boldsymbol{\theta})) = -\mathbb{E}_{\boldsymbol{Y}\sim P_{\boldsymbol{\theta}}} \left[ \frac{\partial}{\partial\boldsymbol{\theta}} E_{\boldsymbol{\theta}}(\boldsymbol{x},\boldsymbol{Y}) \right] \approx -\sum_{j=0}^{k} \frac{\partial}{\partial\boldsymbol{\theta}} E_{\boldsymbol{\theta}}(\boldsymbol{x}, \boldsymbol{Y}_j^-). \tag{4}$$

The resulting approximate NLL objective is equivalent to contrasting the energy value for *real* and *synthesized* targets, with the former being minimized and the latter maximized. The objective is reminiscent of the discriminator's loss in generative adversarial networks (Goodfellow et al., 2014), where a real sample is contrasted to a sample synthesized by the generator network. In practice, setting $k{=}1$ suffices.

The Langevin MCMC algorithm allows for efficient sampling from high dimensional spaces (Geman & Geman, 1984; Neal et al., 2011). Access to the derivative of the unnormalized density function

provides sufficient information for sampling. We apply the following modified transition function and keep only the last sample:

$$\boldsymbol{Y}^{(t+1)} = \boldsymbol{Y}^{(t)} - \frac{\partial E_\theta(\boldsymbol{x}, \boldsymbol{Y}^{(t)})}{\partial \boldsymbol{Y}} + \boldsymbol{U}^{(t)}, \tag{5}$$

with $\boldsymbol{U}^{(t)} \sim \mathcal{N}(0, \epsilon \boldsymbol{I})$, $\epsilon > 0$, $\boldsymbol{Y}^{(0)} \sim \mathcal{N}(0, \epsilon \boldsymbol{I})$ a sample from a fixed initial distribution and $\boldsymbol{Y}^{(T)}$ the final sample. The proper formulation of the Langevin MCMC algorithm multiplies the gradient in Equation 5 by a factor $\epsilon$ and further requires a Metropolis-Hastings acceptance step (Neal, 1993). We forgo both of these components in favor of increased efficiency, but at the cost of forfeiting theoretical guarantees for desirable properties such as not being trapped in a subset of the sampling space, i.e., ergodicity. Discarding all but the last sample $Y^{(T)}$ of each chain constitutes a non typical usage that undermines the usual importance of ergodicity. Notably, this weakens the hard to meet requirement for the sampler to mix between multiple modes in a single MCMC chain, making it sufficient for independently sampled chains to find different local modes. Although the fixed cutoff at $T$ and missing Metropolis-Hastings update result in a biased sampler, previous works have demonstrated the feasibility of training generative models on images with similar Langevin MCMC methods (Xie et al., 2016; 2018; Nijkamp et al., 2020; Du & Mordatch, 2019; Grathwohl et al., 2019).

The model density from Equation 1 approaches the data distribution $P_\mathcal{D}$ while training, leading to an increased ability in distinguishing between synthesized sets $\boldsymbol{Y}^-$ from real sets $\boldsymbol{Y}^+$. This in turn enhances the samples $\boldsymbol{Y}^-$ to be closer to the ground-truth, making it harder for the model to discriminate between real and fake. In practice, it is necessary to smooth out the data distribution. Otherwise, the deep energy-based model would be required to fit a distribution with zero density everywhere except the training examples. Any gradient based sampling and prediction algorithm would be rendered useless. Additional Gaussian distributed noise on the data samples $\boldsymbol{Y}^+$ alleviates this issue and facilitates stable training.

## 2.2 PREDICTION

Prediction from an energy-based viewpoint corresponds to finding the set with the lowest energy value. One approach addresses this intractable optimization problem by approximating a local minimum via gradient descent (Belanger & McCallum, 2016; Belanger et al., 2017). Learning a multimodal distribution is clearly not sufficient, as the deterministic gradient descent algorithm would not be able to cover all possible sets. This would make the learning process pointless, except for a single local minimum in the energy function. We propose to augment the gradient descent optimizer with additional Gaussian noise during the first $n$ steps:

$$\boldsymbol{Y}^{(t+1)} = \boldsymbol{Y}^{(t)} - \frac{\partial}{\partial \boldsymbol{Y}} E_{\boldsymbol{\theta}}(\boldsymbol{x}, \boldsymbol{Y}^{(t)}) + \boldsymbol{U}^{(t)}, \qquad \text{for } t \leq S, \tag{6a}$$

$$\boldsymbol{Y}^{(t+1)} = \boldsymbol{Y}^{(t)} - \frac{\partial}{\partial \boldsymbol{Y}} E_{\boldsymbol{\theta}}(\boldsymbol{x}, \boldsymbol{Y}^{(t)}), \qquad \text{for } S < t \leq T. \tag{6b}$$

For simplicity we choose the same maximum number of steps $T$, both for training and prediction. One interpretation of the prediction procedure is: 1. Langevin MCMC sample $\boldsymbol{Y}^{(S)}$ based on the energy $E_\theta$ and 2. Refine the sample via gradient descent, such that $\boldsymbol{Y}^{(T)}$ is a local minimum of $E_{\boldsymbol{\theta}}$ that is close to $\boldsymbol{Y}^{(S)}$. Note that the partial derivative $\frac{\partial}{\partial \boldsymbol{Y}} E_{\boldsymbol{\theta}}(\boldsymbol{x}, \boldsymbol{Y}^{(t)})$ is not stochastic and can be computed independent of a mini-batch. Thus the sole source of randomness lies with the addition of $U$, resulting in a prediction procedure that allows for different predictions given the same observation.

From the set prediction point of view, the noise term addresses an optimization problem that is specific to set functions. Commonly used set neural networks (Zaheer et al., 2017), require permutation invariant pooling operators. Examples include sum or mean pooling. Both of these result in identical partial gradients for identical elements:

$$\frac{\partial}{\partial \boldsymbol{y}_i} E_{\boldsymbol{\theta}}(\boldsymbol{x}, \boldsymbol{Y}) = \frac{\partial}{\partial \boldsymbol{y}_j} E_{\boldsymbol{\theta}}(\boldsymbol{x}, \boldsymbol{Y}), \tag{7}$$

where $\boldsymbol{y}_i$ and $\boldsymbol{y}_j$ are two different elements in $\boldsymbol{Y}$ with identical value, i.e., $\boldsymbol{y}_i = \boldsymbol{y}_j$. Although we consider set, not multi-set prediction; in practice the set $\boldsymbol{Y}$ needs to be stored as a tensor of numbers

with limited precision. For the purpose of successfully sampling $\boldsymbol{Y}$ from $E_{\boldsymbol{\theta}}$, we restrict the parameters $\boldsymbol{\theta}$ to energy functions with numerically stable derivatives. Specifically, the difference in the gradients of two elements in $\boldsymbol{Y}$ is limited by the difference between the same two elements. This poses the additional difficulty for the optimizer of *separating* different elements that are too close, next to the original task of *moving* the element to the correct position. It is reasonable to assume several elements in close vicinity for problems where the set size is much larger than the number of features. The independently sampled noise term helps disambiguate such proximal elements and speeds up the optimization procedure.

A naive alternative would be to solely initialize the set $\boldsymbol{Y}^{(0)}$ with the constraint of a minimal distance between each element. While this approach addresses the problem at step $t{=}0$, it is ignored in the subsequent steps $t > 0$, where two elements may have collapsed. Our proposed prediction procedure adds independently sampled noise at several steps; thus removing some of the responsibility, for separating elements, from the gradient-based optimizer.

## 3 SET ENERGY

The energy-based viewpoint constitutes an immediate advantage for incorporating symmetry into the neural network architecture. Neural networks that are permutation invariant with respect to the input can be straightforwardly adapted for our purpose. Permutation invariant energy functions have the advantage of being able to define densities directly on sets. Set densities do not require normalization over all possible permutations, as two sequences that are equivalent up to permutation are also equivalent in the sample space. In this section we formulate two different energy functions based on recently proposed permutation invariant neural network architectures that are both compatible with our training and prediction framework.

**Deep Sets**  DeepSets (Zaheer et al., 2017) first applies an MLP on each element, followed by a permutation invariant aggregator and a second MLP. This model is shown to be a universal approximator of continuous permutation invariant functions (Zaheer et al., 2017; Qi et al., 2017). For the set prediction setting, we adopt the following energy function:

$$E_{\mathrm{DS}}(\boldsymbol{x}, \boldsymbol{Y}) = f(\bigoplus_{\boldsymbol{y} \in \boldsymbol{Y}} g([h(\boldsymbol{x}); \boldsymbol{y}])), \tag{8}$$

with $f, g$ denoting MLPs, $h$ a neural network acting on the input, $[\cdot\,;\cdot]$ the concatenation operator and $\bigoplus$ a permutation invariant aggregator. We treat both the observed features and the prediction as input to the neural network, resulting in an energy function that is permutation invariant with respect to the target $\boldsymbol{Y}$.

**Set Encoder**  An alternative set neural network is studied by Zhang et al. (2019). They propose to separately map the observed features and the prediction into a shared latent space. In their case, the distance in the latent space is minimized as a part of the loss function during training. We re-interpret this loss as the energy function:

$$E_{\mathrm{SE}}(\boldsymbol{x}, \boldsymbol{Y}) = L_{\delta}(g(\boldsymbol{Y}) - h(\boldsymbol{x})), \tag{9}$$

with $g$ denoting a permutation invariant neural network, $h$ a neural network acting on the observed features and $L_{\delta}$ the Huber loss. A minimal energy is reached, when both $\boldsymbol{x}$ and $\boldsymbol{Y}$ map to the same point in the latent space. This energy function stands in contrast to $E_{\mathrm{DS}}$, where the observed features directly interact with individual elements in the predicted set.

## 4 RELATED WORK

Our framework is closely related to the works of Belanger & McCallum (2016) and Mordatch (2018), which also take on an energy-based viewpoint. However, they obtain predictions by minimizing the energy via (deterministic) gradient descent and require memory-intensive backpropagation through the unrolled inner optimization during training (Domke, 2012; Belanger et al., 2017). Similarly, deep set prediction network (DSPN) (Zhang et al., 2019) applies the bi-level optimization scheme (Domke, 2012) for learning. Instead of an energy function, DSPN minimizes the distance

between the input and the predicted set in a shared latent space. Our energy-based viewpoint does not require a latent vector space bottleneck between input and prediction, resulting in a broader choice of models. In addition, our prediction algorithm handles multi-modal distributions through additional stochasticity.

Most prior set prediction approaches rely on fixed (Fan et al., 2017) or learned orders (Vinyals et al., 2016). They run into the problem, as identified by Zhang et al. (2020b; 2019), that small changes in the set space may require large changes in the neural network outputs, leading to lower performance. Other approaches require the assumption of independent and identically distributed set elements (Rezatofighi et al., 2017; 2020). Some very recent works (Kosiorek et al., 2020; Carion et al., 2020; Locatello et al., 2020; Karl et al., 2020) respect the permutation symmetry in the model, by applying the Transformer (Vaswani et al., 2017; Lee et al., 2019) without position embedding and a non-autoregressive decoder. Nonetheless, the work of Karl et al. (2020) is limited to set generation. Both Carion et al. (2020) and Locatello et al. (2020) rely on the Hungarian loss as a permutation invariant objective function. Kosiorek et al. (2020) deploy the Chamfer loss augmented with an additional set cardinality objective. By casting learning as conditional density estimation, we forgo the necessity of task specific loss-engineering.

## 5 EXPERIMENTS

The experiments answer two overarching questions: 1. Can our density estimation perspective improve over discriminative training via assignment-based losses? and 2. Can our stochastic prediction algorithm yield multiple plausible sets for multi-modal densities? The experiments also demonstrate the applicability of our approach to a variety of energy functions and a range of set prediction tasks: point-cloud generation, set auto-encoding, object detection and anomaly detection. Code is available at: https://github.com/davzha/DESP.

We investigate the effectiveness of our approach, by comparing against Chamfer and Hungarian loss based training, with predictions formed by deterministic gradient descent. The Chamfer loss assigns every element in the prediction $\hat{Y} = \{y_i\}_{i=1..k}$ to the closest element in the ground-truth $Y$ and vice-versa:

$$L_C(\hat{Y}, Y) = \sum_i \min_j d(\hat{y}_i, y_j) + \sum_j \min_i d(\hat{y}_i, y_j), \tag{10}$$

where $d$ is a vector distance instantiated as the Huber loss in the subsequent experiments. The Hungarian loss is computed by solving the linear assignment problem between the two sets:

$$L_H(\hat{Y}, Y) = \min_{\pi \in S_k} \sum_i d(\hat{y}_i, y_{\pi(i)}), \tag{11}$$

where $S_k$ is the set of all permutations on sets of size $k$. We refer to Appendix A for further details on assignment-based set losses.

### 5.1 COMPUTATIONAL COMPLEXITY ANALYSIS

DESP offers non-trivial computation cost trade-offs, when we compare it to a baseline trained via assignment-based set losses. We identify three main factors that are crucial and specific to our analysis: 1. Number of transition steps $T$, 2. Complexity of the set neural network and 3. Complexity of the loss function. Similar to baselines that form predictions with an inner optimization (Zhang et al., 2019; Belanger & McCallum, 2016), DESP's training and inference time scale linearly with $T$. Though, in practice DESP requires a larger $T$ to achieve reliable sampling quality, potentially resulting in longer training and inference times. The complexity of the set neural network is crucial for determining the computation cost on large set sizes $c$. By choosing a set neural network with time and memory complexity in $\mathcal{O}(c)$, such as DeepSets (Zaheer et al., 2017), DESP can accommodate large set sizes. In comparison to the baselines, DESP avoids the additional computational burden imposed by an assignment-based set loss, which is in $\mathcal{O}(c^2)$ for the Chamfer loss and in $\mathcal{O}(c^3)$ for the Hungarian loss.

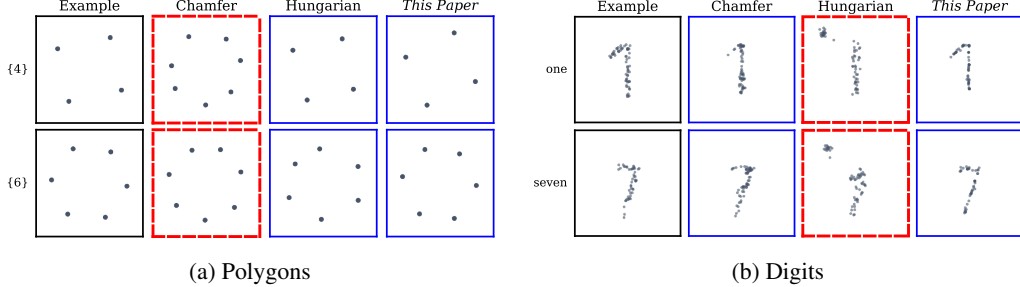

(a) Polygons          (b) Digits

Figure 1: **Data examples and predictions for (a) Polygons and (b) Digits.** Training with Hungarian loss leads to plausible polygons of correct cardinality, while training with Chamfer loss fails to construct a polygon of the desired size. The reverse occurs for Digits, where the Hungarian loss leads to implausible sets. Our method performs favourably on both datasets, by not implicitly interpolating between data examples during training.

## 5.2 GENERATION OF POLYGONS AND DIGITS

Considering set prediction as density estimation can be critical. To illustrate this, we point out fundamental limitations of set loss based training that become apparent when multiple plausible sets exist. In terms of probability densities, each plausible set translates to a local maxima. To study different types of randomness in isolation, we create two synthetic datasets:

- *Polygons* Generate the set of vertices of a convex $n$-sided regular polygon, given the size $x = n$. This task is inherently underdefined; any valid set of vertices can be arbitrarily translated or rotated and remain valid. We limit the scope of permissible answers slightly, by fixing a constant center and radius. Each sample from this dataset has a uniformly randomized angle.

- *Digits* Generate a point-cloud, taking the shape of the digit given by $x$. Point-clouds are sets of low dimensional vectors, describing the spatial arrangement of one or more objects. We limit the digits to $x \in \{\text{one}, \text{seven}\}$, because they are similar and each has different forms, following the most common writing styles (Ifrah, 2000). The shape determines the area from which the set of points $Y$ are sampled. The number of points varies with different shapes, facilitating evaluation of different set sizes and spatial arrangements.

In both datasets the observed feature is kept simple, as our focus lies on predicting sets with non-trivial interrelations. Each example in the dataset consists of a discrete input and a set of 2d vectors as the target, as illustrated in Figure 1. More examples can be found in Appendix B. Both datasets share the notion that several correct sets are permissible, such as an offset in rotation for polygons. The difference between the two datasets lies in the relation that connects different plausible predictions.

**Model** We use the energy function $E_{\text{DS}}$ defined in Equation 8 for both datasets. A 3-layer MLP forms the set equivariant part, followed by a permutation invariant pooling operation and a second 3-layer MLP. We choose FSPool (Zhang et al., 2020b) over more simple aggregators such as sum or mean, as it exhibits much faster convergence rates in preliminary experiments. To accommodate different cardinalities, we zero-pad all sets to a fixed maximum size, similar to Zhang et al. (2019). By ensuring that all non-padding elements are unequal to the zero vector, padding can simply be filtered out from the predictions by setting a threshold around a small area around zero.

**Results** We report the Chamfer loss for the Digits dataset and Hungarian loss for the Polygons dataset in Table 1. The metrics are chosen in a way that aligns with a qualitative assessment (Figure 1) of the performance for each dataset respectively. While the baseline with the Chamfer loss objective performs better on Digits, the Hungarian baseline outperforms the former on Polygons. This result reveals a trade-off when picking set loss functions as training objectives for different types of datasets. Our framework improves over both baselines on both datasets, but more importantly, we do not observe a similar performance discrepancy between the two datasets.

Table 1: **Results for Polygons and Digits** Performance measured in $[10^{-5}]$ as Chamfer loss for Digits and Hungarian loss for Polygons. Lower is better. Training with the Chamfer loss fails on Polygons, while training with the Hungarian loss fails on Digits. The trade-off in the type of randomness that can be handled by an assignment-based set loss does not occur for our approach.

| | Training loss | | Performance on Datasets | |
|---|---|---|---|---|
| | Chamfer | Hungarian | Polygons | Digits |
| Direct risk minimization | ✓ | | $1195.0_{\pm 6.8}$ | $2.4_{\pm 0.3}$ |
| Direct risk minimization | | ✓ | $14.8_{\pm 5.3}$ | $48.0_{\pm 2.3}$ |
| Density estimation (*This paper*) | | | $8.8_{\pm 1.5}$ | $1.8_{\pm 0.1}$ |

We confirm in Figure 1 that both baselines handle the multi-modal data distribution, by interpolating between different target sets. The set loss choice can lead to implausible predictions. While in this case both datasets are designed to be simple for transparency reasons, the choice of set loss becomes a non-trivial trade-off for more complex datasets. Our approach prevents implicit interpolation and consequently does not incur the same trade-off cost. In contrast to purely deterministic optimizers, which always converge to the same local minimum, our stochastic version finds multiple energy minima. Figure 4 and Figure 5 in the appendix demonstrate the ability to cover different modes, where several predictions result in differently rotated polygons and distinctly shaped digits. Each prediction represents an independently sampled trajectory of transitions described in Equation 6. This experiment is tailored towards the special case, when there exist multiple plausible target sets and exemplifies both the short-comings of training with assignment-based set losses and the ability of our approach to predict multiple sets. Whether the results in this simplified experiment will also reflect the superiority of the proposed approach on a real-world problem remains to be tested.

## 5.3 POINT-CLOUD AUTO-ENCODING

Point-cloud auto-encoding maps a variable sized point-cloud to a single fixed-size vector, with the requirement of being able to reconstruct the original point-cloud solely from that vector. Following the setup from Zhang et al. (2019), we convert MNIST (LeCun et al., 2010) into point-clouds, by thresholding pixel values and normalizing the coordinates of the remaining points to lie in $[0, 1]$. We compare against two variations of DSPN (Zhang et al., 2019): 1. Chamfer and 2. Hungarian loss based training. For a fair comparison, we use the energy function $E_{SE}$ defined in Equation 9, with the same padding scheme and hyper-parameters as Zhang et al. (2019). Both baselines average over all intermediate set losses, based on intermediate prediction steps. The padding scheme consists of zero-padding sets to a fixed maximum set size and adding a presence variable for each element, which indicates if the element is part of the set or not. Furthermore, we compare against C-DSPN and TSPN (Kosiorek et al., 2020) on their set size estimation task. They optimize for set size root-mean-squared error (RMSE), in combination with the Chamfer loss. We evaluate our approach based on a single prediction per example.

**Results** Table 2 shows that our approach outperforms C-DSPN and TSPN (Kosiorek et al., 2020) on set size RMSE. We conjecture that this is caused by a conflict between the two objectives: 1. Set size RMSE and 2. Chamfer loss, under limited capacity. While the former requires a cardinality aware representation, the latter does not benefit from a precise cardinality estimation at all. In contrast, our approach does not treat set size as a variable separate from the constituents of the set. Table 3 shows that our approach outperforms both DSPN (Zhang et al., 2019) baselines, even when comparing against the same metric that is used for training the baselines. We explain the increased performance by an ambiguity during reconstruction, induced from the bottleneck in auto-encoding.

Table 2: **Point-cloud auto-encoding** Set size root-mean-square error (RMSE) for set MNIST. Both C-DSPN and TSPN (Kosiorek et al., 2020) first infer the set size, before generating the set. Our approach outperforms both baselines, without explicit set size supervision.

| | Set size RMSE $\downarrow$ |
|---|---|
| C-DSPN | $0.300_{\pm 0.130}$ |
| TSPN | $0.800_{\pm 0.080}$ |
| *This paper* | $0.002_{\pm 0.003}$ |

Table 3: **Point-cloud auto-encoding** Performance measured in Chamfer and Hungarian loss in units of $[10^{-4}]$ for set MNIST. The Chamfer/Chamfer result is from (Zhang et al., 2019), other numbers based on author-provided code. Our approach outperforms both DSPN baselines on both metrics, despite the baselines being directly trained with Chamfer or Hungarian loss.

| | Training loss | | Performance metric | |
|---|---|---|---|---|
| | Chamfer | Hungarian | Chamfer ↓ | Hungarian ↓ |
| DSPN | ✓ | | $0.9_{\pm0.1}$ | $1600.2_{\pm162.7}$ |
| DSPN | | ✓ | $2.8_{\pm0.8}$ | $3.2_{\pm0.8}$ |
| *This paper* | | | $0.6_{\pm0.1}$ | $0.6_{\pm0.1}$ |

As we have observed in the previous experiment, the baselines handle underdefined problems by interpolation in the set space, leading to potentially unrealistic reconstructions. The results indicate that our approach is beneficial even when the underlying data is not explicitly multi-modal. Training the DSPN model with Hungarian loss, instead of Chamfer loss, deteriorates training stability and the reconstructed shape, but captures the set size and point density more faithfully. Augmenting the loss function with set size RMSE alleviates some of the issues with set size, but leads to decreases in shape reconstruction performance (Kosiorek et al., 2020). Our approach does not require any loss-engineering and performs well on all metrics.

**Effect of stochastic prediction**   We study the impact of the proportion of stochastic steps $\frac{S}{T}$, as defined in Equation 6, on the reconstruction performance in Figure 2. Over all runs, the most common minimum energy is approximately at $\frac{S}{T}{=}0.8$. All results reported for our method apply this 0.8 ratio during prediction. Notably, adding stochastic steps improves the energy optimization, in comparison to the fully deterministic gradient descent ($\frac{S}{T}{=}0$). Furthermore, the high correlation between the energy value and the performance leads us to the conclusions that DESP learns more than a simple sampler and that optimization improvements result in increased performance. Our approach is able to produce different plausible predictions at no performance cost.

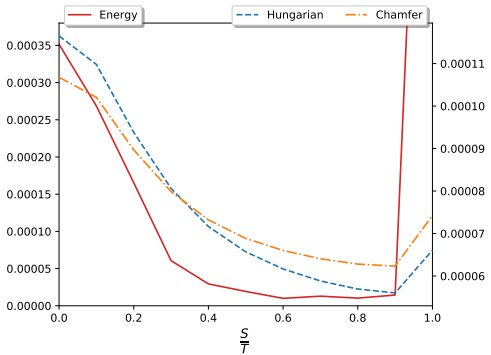

Figure 2: **Effect of stochastic prediction** Test results on set MNIST for different values of $\frac{S}{T}$ (Equation 6). Both energy values (left scale) and set losses (right scale) are optimal, when combining stochastic and deterministic prediction steps. Improvements in the optimizer translate well to increases in performance, as indicated by high correlation between energy and set losses.

## 5.4   Object Set Prediction on CLEVR

In terms of set prediction, object detection tasks the model with learning to predict the set of object locations in an image. Following the previous set prediction literature (Zhang et al., 2019; Kosiorek et al., 2020), we benchmark our method on the CLEVR dataset. While our specific contribution addresses stochastic and underdefined set prediction problems, our method is in principle not limited to those cases. We adopt the same neural network architecture, hyper-parameters and padding scheme as Zhang et al. (2019), to facilitate a fair comparison. The padding scheme is the same as in the previous experiment. The Relation Network (Santoro et al., 2017) in combination with FSPool (Zhang et al., 2020b) takes on the role of the set encoder for $E_{\text{SE}}$, described in Equation 9. We compare against different variations of Chamfer and Hungarian loss based training. Our approach is evaluated based on a single prediction per image.

**Results**   Performance, as seen in Table 4, is measured in average precision (AP) for various intersection-over-union (IoU) thresholds. Similar to the previous experiments, we observe a large discrepancy between training with Chamfer and Hungarian loss. While the Chamfer loss based

Table 4: **Object set prediction on CLEVR** Baselines from Kosiorek et al. (2020) and Zhang et al. (2019) show mixed performance for different intersection-over-union (IoU), whereas our results are consistent and competitive.

| | Training loss | | Performance for different IoU | | | | |
|---|---|---|---|---|---|---|---|
| | Chamfer | Hungarian | $AP_{50}\uparrow$ | $AP_{60}\uparrow$ | $AP_{70}\uparrow$ | $AP_{80}\uparrow$ | $AP_{90}\uparrow$ |
| C-DSPN | ✓ | | $67.7_{\pm5.5}$ | - | - | - | $7.4_{\pm0.9}$ |
| TSPN | ✓ | | $81.2_{\pm1.0}$ | - | - | - | $20.7_{\pm0.2}$ |
| DSPN[†] | | ✓ | $94.0_{\pm0.4}$ | $90.6_{\pm0.6}$ | $82.2_{\pm1.4}$ | $58.9_{\pm3.4}$ | $16.0_{\pm2.4}$ |
| *This paper* | | | $96.2_{\pm0.5}$ | $92.5_{\pm1.0}$ | $80.9_{\pm1.8}$ | $54.3_{\pm3.4}$ | $17.4_{\pm1.6}$ |

[†]The DSPN numbers are from the updated appendix of Zhang et al. (2020a)

training generally outperforms the Hungarian loss for set auto-encoding, the reverse appears to be true for object detection. In comparison, our approach performs consistently on both tasks for all metrics, indicating suitability for general set prediction tasks, beyond multi-modal problems.

## 5.5 SUBSET ANOMALY DETECTION

The objective here is to discover all anomalous faces in a set of images. We re-purpose CelebA (Liu et al., 2015) for subset anomaly detection, by training on randomly sampled sets of size 5 with at least 3 images constituting the inliers, by possessing two or more shared attributes. The set energy function is solely supervised by outlier subset detections, without direct access to attribute values. The challenge during inference lies with implicitly ascertaining the shared attributes, while simultaneously detecting the outliers, including the case where none are present. We examine our method specifically on ambiguous cases, constructed such that different attribute combinations may be considered distinctive for the inliers. Zaheer et al. (2017) consider a similar task, but assume exactly one outlier. Their method can be extended to subsets of variable sizes, by replacing the softmax with a sigmoid (Zaheer et al., 2017), yielding an $F_1$ score of 0.63 on our task. Nonetheless, such an approach is limited to predicting *element-wise probabilities*, which ignores dependencies between individual predictions. Our approach of learning *probabilities over sets* is able to address this challenge, as demonstrated in Figure 3. Given the same set, our method produces multiple valid subset predictions, reflecting an implicit inference of different attribute pairs. This advantage allows DESP to considerably outperform the baseline with an $F_1$ score of **0.76**. Further details can be found in Appendix C.

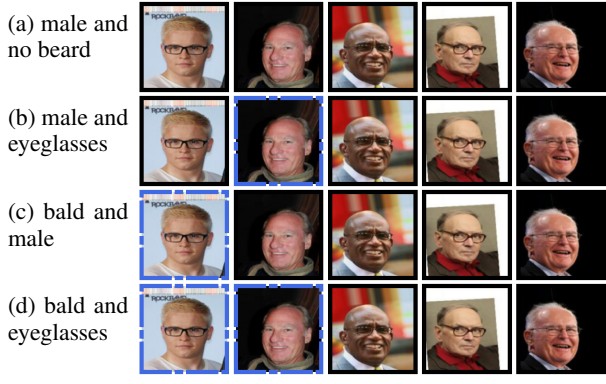

(a) male and no beard

(b) male and eyeglasses

(c) bald and male

(d) bald and eyeglasses

Figure 3: **Subset anomaly detection** Four different outlier subset detections, marked by blue dash-dotted frames, emphasize the ambiguity of what constitutes an anomaly. Inliers share at least two common attributes, e.g., bald and eyeglasses, while outliers lack one or both. Depending on the considered attributes, different subsets are anomalous.

## 6 CONCLUSION

We introduced a new training & prediction framework for set prediction, based on a probabilistic formulation of the task. Our approach addresses the crucial problem of stochastic or underdefined set prediction tasks, where training with assignment-based set losses performs unfavourably. We demonstrated the ability of Deep Energy based Set Prediction (DESP) to learn and predict multiple plausible sets on synthetic data. On non-stochastic benchmarks our method is comparable to previous works, showcasing broad applicability to general set prediction tasks. Finally, we exemplify on the new task of subset anomaly detection the capacity to address tasks beyond those with unambiguous predictions.

ACKNOWLEDGMENTS

This work is part of the research programme Perspectief EDL with project number P16-25 project 3, which is financed by the Dutch Research Council (NWO) domain Applied and Engineering Sciences (TTW).

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

# A    ASSIGNMENT-BASED SET LOSS

Assignment-based set losses compare the predicted set $\hat{Y} = \{\hat{y}_1, \ldots, \hat{y}_k\}$ with the ground-truth set $Y = \{y_1, \ldots, y_l\}$ by element-wise assignments:

$$L_A(\hat{Y}, Y) = \sum_i d(\hat{y}_i, y_{\pi(i)}) + \sum_j d(\hat{y}_{\sigma(j)}, y_j), \tag{12}$$

where $d$ is a vector distance function and $\pi : \{k\} \mapsto \{l\}, \sigma : \{k\} \mapsto \{l\}$ are the assignment functions, which map from one sets' indices to the other. Due to the orderless nature of sets, it is not obvious which element ought to be compared with which. Different assignment-based set losses differ mainly in the assignment strategies, reflected in the choices for $\pi$ and $\sigma$.

The Chamfer loss assigns every element in $\hat{Y}$ to the closest element in $Y$ and vice-versa and can be defined by $\pi(i) = \arg\min_j d(\hat{y}_i, y_j)$ and $\sigma(j) = \arg\min_i d(\hat{y}_i, y_j)$, resulting in:

$$L_C(\hat{Y}, Y) = \sum_i \min_j d(\hat{y}_i, y_j) + \sum_j \min_i d(\hat{y}_i, y_j) \tag{13}$$

For the Hungarian loss $\pi$ and $\sigma$ constitute the inverse functions of each other, thus requiring equal set sizes $n = m$:

$$L_H(\hat{Y}, Y) = \frac{1}{2} \left( \min_{\pi \in S_k} \sum_i d(\hat{y}_i, y_{\pi(i)}) + \min_{\sigma \in S_k} \sum_i d(\hat{y}_{\sigma(j)}, y_j) \right) \tag{14}$$

$$= \min_{\pi \in S_k} \sum_i d(\hat{y}_i, y_{\pi(i)}), \tag{15}$$

where $S_k$ is the set of all permutations on sets of size $k$.

The differences in assignment strategies result in different metric spaces on sets, as illustrated in subsection 5.2. Both the Chamfer and the Hungarian loss exhibit distinct advantages and disadvantages. While the asymptotic compute cost for the Chamfer loss scales in $\mathcal{O}(kl)$ with a set sizes $k, l$, computing the Hungarian loss is much more expensive with a complexity in $\mathcal{O}(k^3)$. The lack of one-to-one assignments for the Chamfer loss, puts it at a disadvantage when comparing multi-sets or sets with multiple similar (up to numerical precision) elements. On the other hand, the strict requirement for bijective assignments for the Hungarian loss disqualifies it when comparing sets of different sizes, i.e., $k \neq l$.

# B    MULTI-MODAL PREDICTIONS

Both Figure 4 and Figure 5 display evidence for the ability of Deep Energy based Set Prediction (DESP) to learn and predict multiple sets for the same input. This ability is important, when we consider datasets with multi-modal target distributions, such as the varying rotation angle for Polygons (Figure 4a) or different writing styles for Digits (Figure 5a). The datasets are described in subsection 5.2.

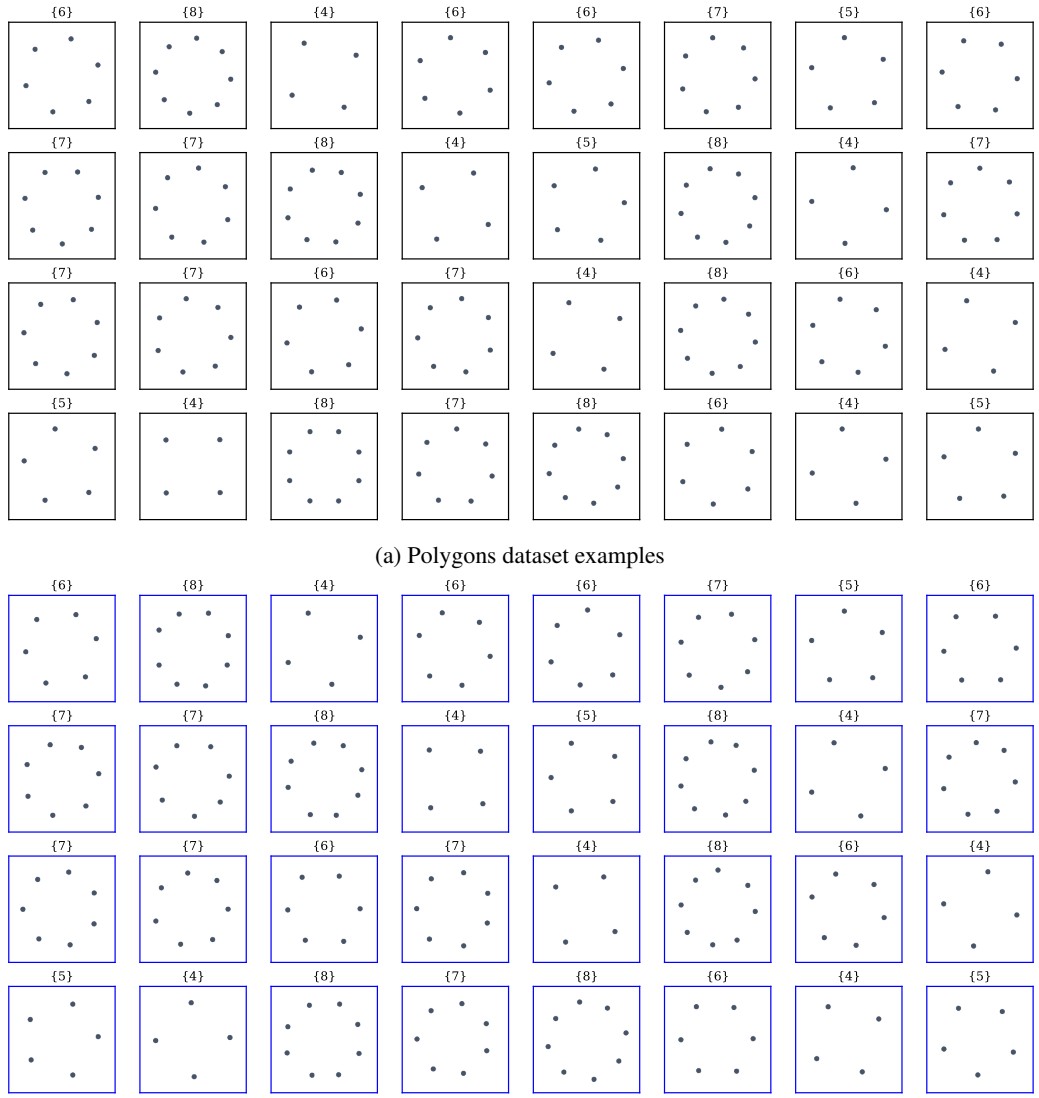

(a) Polygons dataset examples

(b) Model samples by this paper

Figure 4: **Multi-modal Polygons** (a) Polygons of the same cardinality vary in a rotation angle around the center. (b) Our method generates polygons of correct cardinality and estimates varying rotation angles.

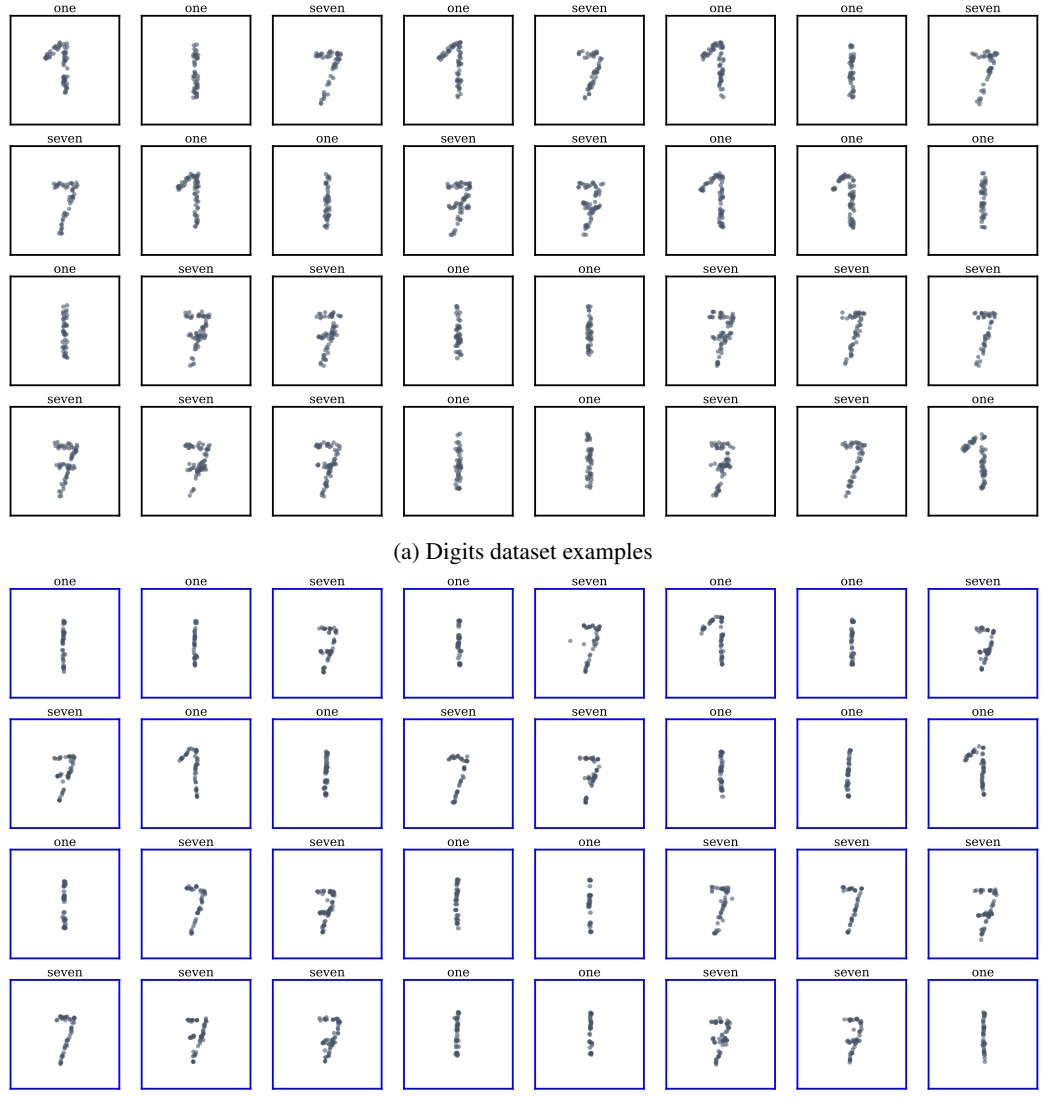

(a) Digits dataset examples

(b) Model samples by this paper

Figure 5: **Multi-modal Digits** (a) The digits one and seven exhibit two different shapes, reflecting differences in writing style. (b) Our method manages to capture both modes for both digits and generate sets from each style respectively.

## C    SUBSET ANOMALY DETECTION

**Dataset preparation**    Individual training instances are sampled from CelebA (Liu et al., 2015) with the following procedure: 1. Sample two attributes $a, b$ 2. Sample 3-5 images that all have attributes $a$ and $b$ 3. Fill the remaining slots with images that do not have both $a$ and $b$ or skip if there are already 5 images. The result of the sampling procedure is an orderless set of 5 images, where the samples from the 2nd and 3rd step constitute the inliers and outliers, respectively. We qualify any subset of images as valid inliers, if they share at least two attributes, which are not possessed simultaneously by any outlier. In order to limit the amount of valid outlier subsets, we restrict the attributes to the following list: Bald, Bangs, Blond Hair, Double Chin, Eyeglasses, Goatee, Gray Hair, Male, No Beard, Wearing Hat, Wearing Necktie. Notably, the training data does not explicitly supervise for attributes and only exposes a single valid subset detection for each training instance.

**Experimental setup**    Each image in the input set is represented by a fixed 128-dimensional feature vector, extracted from the penultimate layer of a ResNet-34 (He et al., 2016) that is optimized for facial attribute detection. We forgo any image augmentation during training and treat the extracted features as a highly informative, albeit flawed, representation of the image. The representation has an accuracy of roughly $\sim 90\%$, as measured by a logistic regression model, for individual attributes and constitutes a source of uncertainty. In addition to each feature vector, we introduce indicator variables $o_i \in \{-1, +1\}$ that constitute the targets and signify if an image is an outlier or not. In order to apply our framework to the discrete targets, we optimize and sample over a convex relaxation of the target domain: $o_i \in [-1, +1]$. We apply an instance of the energy function $E_{\mathrm{DS}}$ (Equation 8) on the set of feature vectors, concatenated with the outlier indicator variables. Both $g$ and $f$ are instantiated as 2-layer MLPs with 256 hidden dimensions. FSPool (Zhang et al., 2020b) is applied as the permutation invariant aggregator. As part of finalizing the predictions, the outlier variables $o_i$ are rounded towards $-1$ or $+1$.

As the baseline, we employ a 4-layer permutation equivariant DeepSets (Zaheer et al., 2017). We use the same extracted features as in our DESP setup as inputs and match the number of parameters. The model is trained with a binary cross-entropy loss that acts on the scalar outputs of the network.

**Evaluation**    The performance is measured on the CelebA test partition images (Liu et al., 2015). Each test example is associated with the full collection of valid subsets, exclusively for evaluation purposes. We consider two distinct test setups: 1. Test instances are sampled the same way as during training, resulting in both unambiguous and ambiguous instances, and 2. Only *ambiguous* instances are used. The first case results in an average number of valid target subsets of $\sim 1.7$, including both ambiguous and unambiguous instances. The second case has an average number of valid target subsets of $\sim 2.3$, with a minimum of 2 valid subset targets.

We measure the proportion of correct predictions per test example as a frequency weighted precision. The frequency of each predicted subset corresponds to the number of appearances across all individual predictions. Recall measures the proportion of all valid subsets that the model manages to predict per test example. We approximate precision and recall in this experiment with 10 predictions, by leveraging the ability of DESP to output multiple subsets. The $F_1$ score is based on the average precision and recall.

**Results**    The performance is reported in Table 5. When solely evaluating on ambiguous cases, the baseline exhibits an $F_1$ performance drop of $\sim 26\%$, as opposed to only $\sim 13\%$ for our method. This highlights the advantage of learning distributions over sets in combination with the ability to produce multiple predictions. Figure 8 and Figure 7 showcase a positive and negative example prediction of our model, respectively. These examples highlight the difficulty of simultaneously determining the shared commonality between the inliers and ascertaining a discrepancy of the outliers.

We examine the effect of varying proportions of stochastic steps in the prediction procedure in Figure 6. Similar to what we observe in the other experiments, $\frac{S}{T}=0.8$ offers the best $F_1$ score. For the fully deterministic case, every predicted set is identical to one another, which is reflected in the low recall score at $\frac{S}{T}=0$.

Table 5: **Subset Anomaly Detection** Performances for the two test setups: 1. Unambiguous + Ambiguous and 2. Ambiguous only. Our method outperforms the baseline, which we derived from DeepSets (Zaheer et al., 2017), on all metrics.

| | Ambiguous | | | Unambiguous + Ambiguous | | |
|---|---|---|---|---|---|---|
| | Precision ↑ | Recall ↑ | $F_1$ ↑ | Precision ↑ | Recall ↑ | $F_1$ ↑ |
| Baseline | $0.75_{\pm 0.02}$ | $0.34_{\pm 0.01}$ | $0.47_{\pm 0.01}$ | $0.73_{\pm 0.02}$ | $0.56_{\pm 0.01}$ | $0.63_{\pm 0.01}$ |
| *This paper* | $0.76_{\pm 0.02}$ | $0.58_{\pm 0.01}$ | $0.66_{\pm 0.00}$ | $0.74_{\pm 0.02}$ | $0.78_{\pm 0.01}$ | $0.76_{\pm 0.01}$ |

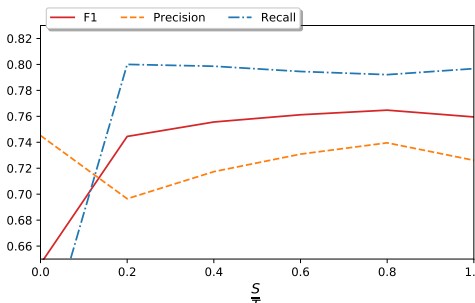

Figure 6: **Anomaly detection ablation** Test results on subset anomaly detection for different values of $\frac{S}{T}$ (Equation 6). Inclusion of stochasticity in the prediction procedure significantly improves the recall performance compared to the fully deterministic case ($\frac{S}{T}=0$). The $F_1$ score is highest at $\frac{S}{T}=0.8$, representing the best trade-off point between recall and precision.

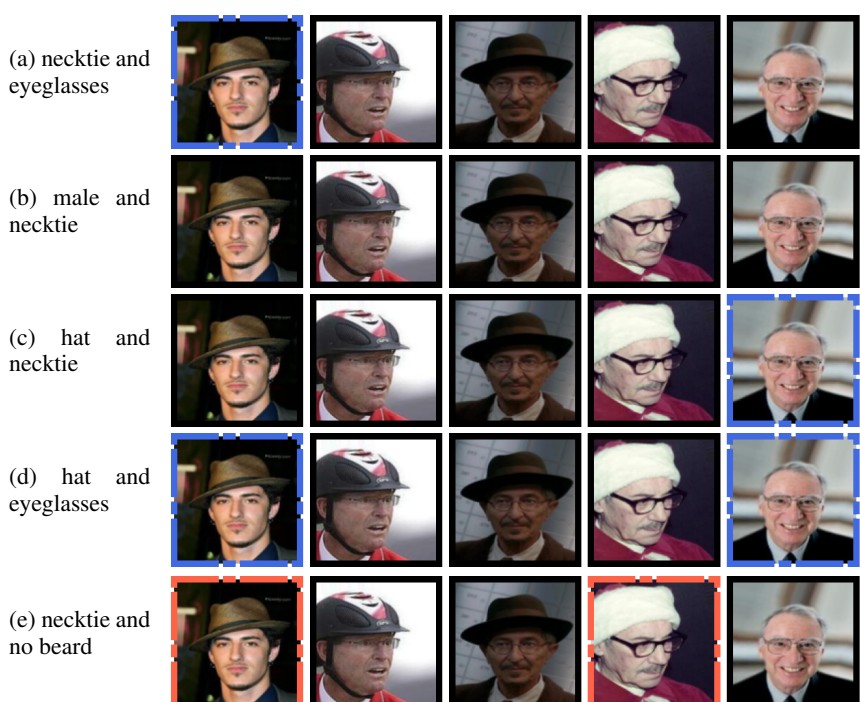

(a) necktie and eyeglasses

(b) male and necktie

(c) hat and necktie

(d) hat and eyeglasses

(e) necktie and no beard

Figure 7: **Subset anomaly detection negative example** (a)-(d) Four correct outlier subset detections, marked by blue dash-dotted frames, predicted by the model. The subset (e), marked by red dash-dotted frames, constitutes and error, because it is a valid prediction, that is missed by the model. Multiple subset possibilities showcase how challenging the subset anomaly detection task is.

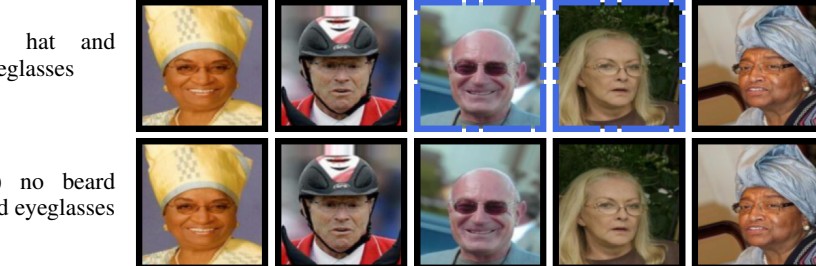

(a) hat and eyeglasses

(b) no beard and eyeglasses

Figure 8: **Subset anomaly detection positive example** Two different outlier subset detections marked by blue dash-dotted frames predicted by the model. While the inliers in both rows share the *no beard* attribute, (a) reflects the case where the defining feature instead consists of wearing a *hat* and *eyeglasses*. The empty subset prediction in (b) reflects a case, where the model correctly predicts the absence of outliers.

