# OpenReview forum: "Set Prediction without Imposing Structure as Conditional Density Estimation"
_ICLR.cc/2021/Conference — ICLR 2021 Poster_

### Official Review · AnonReviewer2 · 2020-10-28
**A novel framework for set prediction**

**Rating:** 7
**Confidence:** 3

**Review:**

Authors propose a new method for formulating set prediction tasks. They propose to use a noisy energy-based model with langevin mcmc + noisy startup as their model. The can approximate the gradient of the likelihood function by computing the enery of ground truth pairs and energy of synthesized pairs where the target is sampled from the model distribution.

A major advantage of this formulation with contrastive GAN style loss (competing synthesized vs gt and improving synthesized) over the previous related work which optimized a distance metric suitable for sets is that it works on a wider range of tasks. Experiments show that most common metrics (hungarian and chamfer distance) both fail in certain tasks where the energy based contrastive approach prevails. Apparently they only need 1 synthetic set to approximate the gradient of partition function which is surprising. Given this, one expects their computation cost and training time to be on par with metric based methods. Unfortunately such analysis is currently missing.

One of their main contributions is advocating for adding gaussian noise to the first 80% of the steps. They reason it is an effective way of covering multi modal scenarios. They achieve impressive results on anomaly detection tasks which they attribute mainly to their multi modal abilities. It would be interesting to have an ablation on the anomaly detection task with no start up noise in the mcmc algorithm.

Another question that arises from their stochastic generation is regarding mnist autoencoding and clevr task. It is not clear if they compute the results based on how many samples per example.

Pros: The paper is relatively well written.
They cover several different tasks in their experimental analysis.
The merits of optimizing for a distribution over the sets is explained well, is intuitive and experiments shows that it works.
They introduce several novel ideas that can be significant in this line of research later on.

cons:
lack of computation cost, training time, inference time analysis. It is not clear if this method scales to larger tasks with many more elements in the set. For example they can tackle shapenet point cloud reconstruction as a larger scale dataset.
Their datasets are toy scale in general, but given that their method consistently outperforms previous work is not a major shortcoming.

niit: The overloading of variable Z is confusing. In page 2, Z is both the partitioning function and a random noise added to the transition function for smoothing the gradients.

--------------------Post Author Response
Thank you for addressing my concern about complexity and adding Fig. 6. It seems that it is still better than baselines in terms of complexity.

---

> ### Author Response · Authors · 2020-11-20
> **Response for AnonReviewer2**
>
> We thank AnonReviewer2 for the detailed and constructive feedback.
>
> **Apparently they only need 1 synthetic set to approximate the gradient of partition function which is surprising. Given this, one expects their computation cost and training time to be on par with metric based methods. Unfortunately such analysis is currently missing.**
>
> We address the computation cost and training time in the first part of section 5.1 of the revised paper:
> "DESP offers non-trivial computation cost trade-offs, when we compare it to a baseline trained via assignment-based set losses. We identify three main factors that are crucial and specific to our analysis: 1. Number of transition steps $T$, 2. Complexity of the set neural network and 3. Complexity of the loss function. Similar to baselines that form predictions with an inner optimization (Zhang et al. 2019; Belanger et al. 2017), DESP’s training & inference time scale linearly with $T$. Though, in practice DESP requires a larger $T$ to achieve reliable sampling quality, potentially resulting in longer training and inference times."
>
> **It would be interesting to have an ablation on the anomaly detection task with no start up noise in the mcmc algorithm.**
>
> Indeed, an ablation over the anomaly detection task offers additional insight into the prediction algorithm. Removing the start up noise has little effect on the performance ($<0.01$ differences). The prediction procedure finds an easy substitute for the missing initial noise in the subsequent noise samples $U^{(1)}, U^{(2)}, \ldots$.
> We added figure 6 to the result discussion of the subset anomaly detection experiment in appendix C, accompanied by the following paragraph:
> "We examine the effect of varying proportions of stochastic steps in the prediction procedure in figure 6. Similar to what we observe in the other experiments, $\frac{S}{T}=0.8$ offers the best $F_1$ score. For the fully deterministic case, every predicted set is identical to one another, which is reflected in the low recall score at $\frac{S}{T}=0$."
>
> **Another question that arises from their stochastic generation is regarding mnist autoencoding and clevr task. It is not clear if they compute the results based on how many samples per example.**
>
> The revised paper is updated with the following clarifications:
> In section 5.3, Point-Cloud Auto-Encoding: “We evaluate our approach based on a single prediction per example.”
> In section 5.4, Object Set Prediction on CLEVR: “Our approach is evaluated based on a single prediction per image.”
>
> **lack of computation cost, training time, inference time analysis. It is not clear if this method scales to larger tasks with many more elements in the set. For example they can tackle shapenet point cloud reconstruction as a larger scale dataset.**
>
> We address the scaling to large set sizes in the second part of section 5.1 of the revised paper:
> "The complexity of the set neural network is crucial for determining the computation cost on large set sizes $c$. By choosing a set neural network with time and memory complexity in $\mathcal{O}(c)$, such as DeepSets (Zaheer et al. 2017), DESP can accommodate large set sizes. In comparison to the baselines, DESP avoids the additional computational burden imposed by an assignment-based set loss, which is in $\mathcal{O}(c^2)$ for the Chamfer loss and in $\mathcal{O}(c^3)$ for the Hungarian loss."
>
> **The overloading of variable $\mathbf{Z}$ is confusing. In page 2, $\mathbf{Z}$ is both the partitioning function and a random noise added to the transition function for smoothing the gradients.**
>
> Thank you for pointing out the overloading of the variable $\mathbf{Z}$. We fix this in the updated paper, by introducing the symbol $\mathbf{U}^{(t)}$ for the noise in equation 5 and 6. The additional superscript $(t)$ clarifies that it is sampled for each step.

---

### Official Review · AnonReviewer4 · 2020-10-28

**Rating:** 7
**Confidence:** 3

**Review:**

This paper poses set prediction as a conditional density estimation problem and subsequently develops an energy-based model training/inference procedure. The motivation for this approach is twofold: existing approaches which impose structure via loss functions can induce model bias based on the metric chosen, and existing approaches also induce bias in the sense that they cannot learn multimodal distributions of outputs when, for example, we may want to observe and rank various candidate sets for a given input. These motivations clearly frame the development of the methods in this paper, and the experiments do a good job recalling these motivations as the focus for comparison to previous approaches. Examples in multiple domains are shown to support the hypothesis that this paper's approach performs better than existing approaches. Overall, the writing is clear and the paper flows well. I think overall the paper combines techniques from various fields, namely energy-based methods and set prediction, that renders it useful for the community.

Ideally, I would have liked to have seen some theoretical results based on a simplified setup, indicating more rigorously the ability of this approach in provably adapting to ill-conditioned metrics and learning multimodal distributions. A variety of results for Langevin MCMC exist that can be exploited to provide this analysis. In its current form, I think the paper is acceptable for submission and is relevant to the community, but some more theoretical analysis will definitely add value to a portion of readers like myself.

Some questions/feedback for the authors:
I think there may be a negative sign missing from Equation 4. Also, it may be useful to further explain why contrasting energy for real and synthesized examples makes sense. I know this is a common setup for structured prediction problems, but I think it is worth paying some lip service in the text. Furthermore, I found it somewhat odd that Equation 5 was framed as an instantiation of Langevin MCMC (perhaps it may be more useful to frame it as an instance of SGLD?), but not equation 6 (at least 6a). This also introduces a natural follow up of whether doing Hamiltonian Monte Carlo based methods makes any difference in mixing efficiency on the experimental setups. Finally, Table 1 was somewhat confusing. The caption explains that Chamfer is bad on Polygons, and Hungarian is bad for Digits, but the numbers don't seem to follow that description. Maybe the rows or columns got switched?

---

> ### Author Response · Authors · 2020-11-20
> **Response for AnonReviewer4**
>
> We thank AnonReviewer4 for the detailed and actionable feedback.
>
> **Ideally, I would have liked to have seen some theoretical results based on a simplified setup, indicating more rigorously the ability of this approach in provably adapting to ill-conditioned metrics and learning multimodal distributions. A variety of results for Langevin MCMC exist that can be exploited to provide this analysis. In its current form, I think the paper is acceptable for submission and is relevant to the community, but some more theoretical analysis will definitely add value to a portion of readers like myself.**
>
> We added the following paragraph in section 2.1 to the revised paper:
> "The proper formulation of the Langevin MCMC algorithm multiplies the gradient in equation 5 by a factor $\epsilon$ and further requires a Metropolis-Hastings acceptance step (Neal et al. 1993). We forgo both of these components in favor of increased efficiency, but at the cost of forfeiting theoretical guarantees for desirable properties such as not being trapped in a subset of the sampling space, i.e. ergodicity. Discarding all but the last sample $Y^{(T)}$ of each chain constitutes a non typical usage that undermines the usual importance of ergodicity. Notably, this weakens the hard to meet requirement for the sampler to mix between multiple modes in a single MCMC chain, making it sufficient for multiple independently sampled chains to find different local modes. Although the fixed cutoff at $T$ and missing Metropolis-Hastings update result in a biased sampler, previous works have demonstrated the feasibility of training generative models on images with similar Langevin MCMC methods (Xie et al., 2016; 2018; Nijkamp et al., 2020; Du & Mordatch, 2019; Grathwohlet al., 2019)."
>
> **Some questions/feedback for the authors: I think there may be a negative sign missing from Equation 4.**
>
> Thank you for pointing out the missing negative sign, which we fixed in the updated paper.
>
> **Also, it may be useful to further explain why contrasting energy for real and synthesized examples makes sense. I know this is a common setup for structured prediction problems, but I think it is worth paying some lip service in the text.**
>
> In order to provide a better intuition behind the training objective, we added the following sentence in section 2.1:
> “The objective is reminiscent of the discriminator’s loss in generative adversarial networks (Goodfellow et al., 2014), where a real sample is contrasted to a sample synthesized by the generator network.”
>
> **Furthermore, I found it somewhat odd that Equation 5 was framed as an instantiation of Langevin MCMC (perhaps it may be more useful to frame it as an instance of SGLD?), but not equation 6 (at least 6a).**
>
> We suspect that the usage of the term Stochastic Gradient Langevin Dynamics (SGLD) (Welling & Teh, 2011) is inconsistent in the literature. Following the original work (Welling & Teh, 2011), we understand SGLD to refer to the specific case when the gradient is approximated by a stochastic version, e.g. using mini-batches. Our usage of the langevin dynamics does not require stochastic gradients, motivating our preference for the term Langevin MCMC.
> We added a clarification after equation 6:
> “Note that the partial derivative $\frac{\partial}{\partial \mathbf{Y}}E_\mathbf{\theta}(\mathbf{x},\mathbf{Y}^{(t)})$ is not stochastic and can be computed independent of a mini-batch.”
> We agree with the reviewer that framing 6a as Langevin MCMC as well improves the clarity and it is included in the revision after the equation 6:
> “One interpretation of the prediction procedure is: 1. Langevin MCMC sample $\mathbf{Y}^{(S)}$ based on the energy $E_\mathbf{\theta}$ and 2. Refine the sample via gradient descent, such that $\mathbf{Y}^{(T)}$ is a local minimum of $E_\mathbf{\theta}$ that is close to $\mathbf{Y}^{(S)}$.”
>
> **This also introduces a natural follow up of whether doing Hamiltonian Monte Carlo based methods makes any difference in mixing efficiency on the experimental setups.**
>
> Preliminary tests on including momentum during sampling (Hamiltonion Monte Carlo) destabilized training. While it appears plausible that Hamiltonion Monte Carlo should improve sampling efficiency, it is currently unclear if it is desirable from a training perspective. More specifically, it introduces larger changes to the state space, which we speculate may be the cause for destabilizing training. We believe further examination into this topic would be of great interest, but see it as outside the scope of the current paper.
>
> **Finally, Table 1 was somewhat confusing. The caption explains that Chamfer is bad on Polygons, and Hungarian is bad for Digits, but the numbers don't seem to follow that description. Maybe the rows or columns got switched?**
>
> Thank you for pointing out the error in Table 1. Indeed, the column names were mixed up and we fixed the table in the updated paper.

---

### Official Review · AnonReviewer1 · 2020-10-28
**Conditional density estimation and noisy optimization for set prediction**

**Rating:** 6
**Confidence:** 2

**Review:**

## Summary
This paper introduces a two step process to learning models for set prediction tasks (such as identifying locations of objects in images, generating polygons or point clouds).

The first phase learns an energy model of the probability of a given set Y given features x, modeled using deep networks. This allows to use the negative log-likelihood as a loss rather than assignment based losses, which allows to model multiple plausible sets given a specific choice of features. To learn this model, the authors approximate the NLL by sampling from data, and, when necessary, generating synthetic sets Y from the current model iteration.

In the set prediction phase, the learned energy model is optimized using noisy SGD during the first few iterations in order to potentially sample from multiple different optima.

## Quality
This paper is well motivated, and the experiments clearly show the benefits of avoiding assignment-based loss functions. However, some statements and design choices would, in my opinion, have benefited from deeper theoretical or empirical justifications.

My main question is regarding the use of the noise Z in section 2.2. The noise is motivated by the discovery of multiple optima. Do the authors have any results showing that this is indeed the downstream effect of the noise (e.g., showing a greater diversity in generated polygons)? Noisy SGD is also known to help with non-convex optimization problems, regardless of identifying multiple minima.

## Clarity
Overall this paper was clear, but several clarifications could be added to improve readability, such as providing the mathematical form of the Hungarian and Chamfer losses, and avoiding the use of $Z$ for both the partition function $Z(x; \theta)$ and the random variable $\sim \mathcal N(0, \epsilon I)$.

I am also curious to know how the sets are encoded: previous work (e.g., Belanger et al.) looked at tasks where the ground set is finite, and the zero-one encoding of the set is made continuous for the purpose of the optimization, then projected back into discrete space. However, given the chosen tasks in this paper, it seems like the encodings are simply continuous coordinates, with the cardinality being constrained. Is this correct?

Finally, is there a minus sign missing from the derivation at Eq. (4)?

## Originality
My understanding is that this paper proposes two key contributions:
 - learning the conditional energy density by sampling and generating synthetic examples
 - using noisy SGD steps for the first $S$ iterations during prediction.

If my understanding is correct, I would have like to see these two contributions explored in more detail. The learning procedure seems similar to noise contrastive estimation (but with a more subtle definition of "noisy samples"). Is this point of view correct? If so, did you explore other noise distributions for the synthetic samples?

As mentioned above, I would also be curious to know if the noisy SGD during estimation leads to improved diverse mode sampling, as well.

# Significance
Modeling distributions over sets with neural networks is, as the authors point out, a difficult problem due in part to  permutation invariance. The use of the NLL under a distribution over sets as a loss function is an elegant way of avoiding this problem, and generalizes approaches that already assume a more specific energy form. The significance of this work lies in how this NLL is optimized, and how good samples are then drawn from the learned distribution. However, I believe that a more thorough investigation of how the proposed method compares to other methods to learn the energy density or optimize the sampling process would increase this paper's impact.

---

> ### Author Response · Authors · 2020-11-20
> **Response for AnonReviewer1**
>
> We thank AnonReviewer1 for the thorough and helpful feedback.
>
> **In the set prediction phase, the learned energy model is optimized using noisy SGD during the first few iterations in order to potentially sample from multiple different optima. [...] My main question is regarding the use of the noise Z in section 2.2. The noise is motivated by the discovery of multiple optima. Do the authors have any results showing that this is indeed the downstream effect of the noise (e.g., showing a greater diversity in generated polygons)?**
>
> We clarify that during both the first and second phase, we do not require stochastic gradients when sampling or optimizing for $\mathbf{Y}$. We added in section 2.2:
> “Note that the partial derivative $\frac{\partial}{\partial \mathbf{Y}}E_\mathbf{\theta}(\mathbf{x},\mathbf{Y}^{(t)})$ is not stochastic and can be computed independent of a mini-batch. Thus the sole source of randomness lies with the addition of the noise $\mathbf{U}^{(t)}$, resulting in a prediction procedure that allows for multiple different predictions.”
> Furthermore, we added figure 6 to the result discussion of the subset anomaly detection experiment in appendix C, accompanied by the following paragraph:
> "We examine the effect of varying proportions of stochastic steps in the prediction procedure in figure 6. Similar to what we observe in the other experiments, $\frac{S}{T}=0.8$ offers the best $F_1$ score. For the fully deterministic case, every predicted set is identical to one another, which is reflected in the low recall score at $\frac{S}{T}=0$."
>
> **Overall this paper was clear, but several clarifications could be added to improve readability, such as providing the mathematical form of the Hungarian and Chamfer losses, and avoiding the use of Z for both the partition function Z(x;θ) and the random variable ∼N(0,ϵI).**
>
> We add the suggested formulation of the set losses to section 5, supplemented by further discussion in Appendix A. We fix double usage of $\mathbf{Z}$, by introducing $\mathbf{U}^{(t)}$ for the noise in equation 5 and 6. The additional superscript $(t)$ clarifies it is sampled for each step. Thank you.
>
> **I am also curious to know how the sets are encoded: previous work (e.g., Belanger et al.) looked at tasks where the ground set is finite, and the zero-one encoding of the set is made continuous for the purpose of the optimization, then projected back into discrete space. However, given the chosen tasks in this paper, it seems like the encodings are simply continuous coordinates, with the cardinality being constrained. Is this correct?**
>
> Indeed, we adopt the same strategy as Belanger et al. (2017) for discrete variables. We clarify the subset anomaly detection task in appendix C:
> “In order to apply our framework to the discrete targets, we optimize and sample over a convex relaxation of the target domain: $o_i\in[-1,+1]$.”
> We clarify in section 5.2:
> “To accommodate different cardinalities, we zero-pad all sets to a fixed maximum size, similar to Zhang et al. (2019). By ensuring that all non-padding elements are unequal to the zero vector, padding can simply be filtered out from the predictions by setting a threshold around a small area around zero.“.
> And in section 5.3, we add:
> “The padding scheme consists of zero-padding sets to a fixed maximum set size and adding a presence variable for each element, which indicates if the element is part of the set or not.” and refer back to the clarification in section 5.4.
>
> **Finally, is there a minus sign missing from the derivation at Eq. (4)?**
>
> Thank you, fixed.
>
> **My understanding is that this paper proposes two key contributions:**
> **- learning the conditional energy density by sampling and generating synthetic examples**
> **- using noisy SGD steps for the first  iterations during prediction.**
> **If my understanding is correct, I would have like to see these two contributions explored in more detail. The learning procedure seems similar to noise contrastive estimation (but with a more subtle definition of "noisy samples"). Is this point of view correct? If so, did you explore other noise distributions for the synthetic samples?**
>
> We examine the effect of noisy steps during the prediction procedure in figure 2, where the fraction of noisy steps is varied from $0\%$ to $100\%$. As noted above, discovering multiple modes is not possible for the fully deterministic baselines. Approximating the gradient of the partition function via sampling has been applied earlier (e.g. in [1]) and we do not consider this constituent of the framework to be a key contribution in itself. Instead we would like to emphasize that the full training & prediction framework constitutes our main contribution. It manages to remove the limitations of assignment-based set loss training, such as the imposed structure, which we exemplify in a diverse range of experiments.
>
> [1] Teh, Y., et al. Energy-based models for sparse overcomplete representations. JMLR, 4:1235–1260, 2003.

---

### Official Review · AnonReviewer3 · 2020-10-29
**unsubstantiated claims, simplified experiments and unclear evaluation**

**Rating:** 6
**Confidence:** 3

**Review:**

The paper concerns a learning framework to predict set by formulating it as a conditional density estimation problem. The approach relies on deep energy-based models and predicts multiple plausible sets using gradient-guided sampling. The suggested method has been evaluated on a variety of set prediction problems with competitive performance compared to few set prediction baselines.

The idea of predicting set using energy-based models seems to be interesting. It is also important to predict but more than one plausible set reflecting mode/sample diversity in the underlying set distribution of the data.

However, my major concerns about the submission are:

1- Few unsubstantiated claims:
a) all over the text it is claimed that the proposed model can capture and learn multi-modal set densities and I am not sure how this is possible with the current simplified sampling strategy (which use a perturbation of gradient similar to cheap MCMC with SGD) with no theoretical guarantee can capture this complex underlying distribution.
b) It is also inaccurate to claim that multi-modal distribution over sets cannot be learned directly from a set loss (a sentence in introduction). The definition of set loss is unclear in this context, but the losses can be derived by modelling a set distribution (possibly multi-modal) parametrically where the parameters of this distribution can be learned using deep neural network


 2- diverse but simplified experiments and unclear evaluations:
a) The experiments are diverse enough, but some of the setups are very simplified (eg generation of polygons & Digits experiments which considers a perfect input x and two numbers only). These simplifications might not necessarily reflect the superiority of the proposed approach compared to the baselines when tested in a real-world problem.
CLEVER for the task of object detection also seems to be very simple dataset. But even with this, the proposed approach does not seem to be superior compared to DSPN under specific IoU thresholds, why is this the case?

b) The suggested approach is claimed to generate multiple plausible sets, are the samples derived from diverse modes or it is an importance sampling? I cannot find out these from the samples in the Figures provided in the appendix.  how are these samples weighted and which sample is used for the valuations?

---

> ### Author Response · Authors · 2020-11-20
> **Response for AnonReviewer3**
>
> We thank AnonReviewer3 for the detailed and constructive comments.
>
> **a) all over the text it is claimed that the proposed model can capture and learn multi-modal set densities and I am not sure how this is possible with the current simplified sampling strategy (which use a perturbation of gradient similar to cheap MCMC with SGD) with no theoretical guarantee can capture this complex underlying distribution.**
>
> We clarify in section 2.1 of the revised paper:
> “The proper formulation of the Langevin MCMC algorithm multiplies the gradient in equation 5 by a factor $\epsilon$ and further requires a Metropolis-Hastings acceptance step (Neal et al. 1993). We forgo both of these components in favor of increased efficiency, but at the cost of forfeiting theoretical guarantees for desirable properties such as not being trapped in a subset of the sampling space, i.e. ergodicity. Discarding all but the last sample $Y^{(T)}$ of each chain constitutes a non typical usage that undermines the usual importance of ergodicity. Notably, this weakens the hard to meet requirement for the sampler to mix between multiple modes in a single MCMC chain, making it sufficient for multiple independently sampled chains to find different local modes. Although the fixed cutoff at $T$ and the missing Metropolis-Hastings update result in a biased sampler, previous works have demonstrated the feasibility of training generative models on images with similar Langevin MCMC methods (Xie et al., 2016; 2018; Nijkamp et al., 2020; Du & Mordatch, 2019; Grathwohlet al., 2019).”
>
> **b) It is also inaccurate to claim that multi-modal distribution over sets cannot be learned directly from a set loss (a sentence in introduction). The definition of set loss is unclear in this context, but the losses can be derived by modelling a set distribution (possibly multi-modal) parametrically where the parameters of this distribution can be learned using deep neural network**
>
> We acknowledge the ambiguity of the set loss expression used during the introduction and extended the revised paper with precise mathematical formulations in section 5, further supplemented by appendix A. Specifically, we consider assignment-based set losses, such as the Chamfer or Hungarian loss used by (Zhang et al., 2019), which are typically applied in set prediction and cannot be used to learn multi-modal set densities with deep neural networks.
>
> **2- diverse but simplified experiments and unclear evaluations: a) The experiments are diverse enough, but some of the setups are very simplified (eg generation of polygons & Digits experiments which considers a perfect input x and two numbers only). These simplifications might not necessarily reflect the superiority of the proposed approach compared to the baselines when tested in a real-world problem.**
>
> The reviewer is right. We added the following sentences to section 5.2:
> “This experiment is tailored towards the special case, when there exist multiple plausible target sets and exemplifies both the short-comings of training with assignment-based set losses and the ability of our approach to predict multiple sets. Whether the results in this simplified experiment will also reflect the superiority of the proposed approach on a real-world problem remains to be tested.”
>
> **CLEVER for the task of object detection also seems to be very simple dataset. But even with this, the proposed approach does not seem to be superior compared to DSPN under specific IoU thresholds, why is this the case?**
>
> The CLEVR object detection experiment does not represent the type of problems that motivate our proposed framework. Our goal is to address stochastic or underdefined set prediction tasks. The experiment serves to validate and compare our method to previous approaches. Notably, we apply the same neural network and hyper-parameters as DSPN. The results show that our method is generally on-par with DSPN, despite no additional fine-tuning towards our proposed training scheme. The fact that our framework does not outperform the baselines at every IoU threshold, does not detract from the main thesis of our work. Namely, addressing tasks with multiple valid targets, which is supported by the other three experiments.
>
> **b) The suggested approach is claimed to generate multiple plausible sets, are the samples derived from diverse modes or it is an importance sampling? I cannot find out these from the samples in the Figures provided in the appendix. how are these samples weighted and which sample is used for the valuations?**
>
> The revised paper is updated with the following clarifications:
> In section 5.2, Generation of Polygons & Digits: “Each prediction represents an independently sampled trajectory of transitions described in Equation 6.”
> In section 5.3, Point-Cloud Auto-Encoding: “We evaluate our approach based on a single prediction per example.”
> In section 5.4, Object Set Prediction on CLEVR: “Our approach is evaluated based on a single prediction per image.”

---

### Decision · Program_Chairs · 2021-01-07
**Final Decision**

**Decision:**

Accept (Poster)

**Comment:**

The paper proposes to predict sets using conditional density
estimates. The conditional densities of the reponse set given the
observed features is modeled through an energy based function. The
energy function can be specified using tailored neural nets like deep
sets and is trained trough approximate negative log likelihoods using
sampling.

The paper was nice to read and was liked by all the reviewers. The one
thing that stood out to me was the emphasis on multi-modality. (multi
appears 51 times).  This could be toned down because little is said
about the quality relative to the true p(Y | x) and the focus is
mainly on the lack of this in existing work.